# Design and Fabrication of a Ka Band RF MEMS Switch with High Capacitance Ratio and Low Actuation Voltage

**DOI:** 10.3390/mi13010037

**Published:** 2021-12-28

**Authors:** Kun Deng, Fuxing Yang, Yucheng Wang, Chengqi Lai, Ke Han

**Affiliations:** 1School of Automation, Beijing University of Posts and Telecommunications, Haidian District, Beijing 100876, China; yangfx@bupt.edu.cn; 2School of Electronic Engineering, Beijing University of Posts and Telecommunications, Haidian District, Beijing 100876, China; wangyucheng@bupt.edu.cn (Y.W.); laichengqi@bupt.edu.cn (C.L.); hanke@bupt.edu.cn (K.H.)

**Keywords:** RF MEMS switch, low voltage, capacitance radio, MIM capacitors

## Abstract

In this paper a high capacitance ratio and low actuation voltage RF MEMS switch is designed and fabricated for Ka band RF front-ends application. The metal-insulator-metal (MIM) capacitors is employed on a signal line to improve the capacitance ratio, which will not degrade the switch reliability. To reduce the actuation voltage, a low spring constant bending folding beam and bilateral drop-down electrodes are designed in the MEMS switch. The paper analyzes the switch pull-in model and deduces the elastic coefficient calculation equation, which is consistent with the simulation results. The measured results indicated that, for the proposed MEMS switch with a gap of 2 μm, the insertion loss is better than −0.5 dB and the isolation is more than −20 dB from 25 to 35 GHz with an actuation voltage of 15.8 V. From the fitted results, the up-state capacitance is 6.5 fF, down-state capacitance is 4.3 pF, and capacitance ratios is 162. Compared with traditional MEMS capacitive switches with dielectric material Si_3_N_4_, the proposed MEMS switch exhibits high on/off capacitance ratios of 162 and low actuation voltage.

## 1. Introduction

In today’s competitive wireless market, the compact, low cost, reconfigurable and multi-function radio frequency (RF) blocks is the main research focus. The radio frequency micro-electromechanical systems (RF MEMS) switch draws considerable attention owing to its attractive performance, as a key enabler for reconfigurable RF front-ends [1,2]. Compared to PIN diode or FET switches, an RF MEMS switch possess many advantages, such as high isolation, low insertion loss and low or near-zero power consumption [3], and it is widely applied in reconfigurable antenna, tunable filter, phase shifting network and automatic test equipment [4,5].

There are two important indices to measure the switch performance, Capacitance ratio and actuation voltage. A high capacitance ratio is beneficial to achieve high isolation and excellent RF performance, for example, in the application of the tunable filter, the capacitance ratio of the switch determines the adjustable range of the center frequency of the resonant unit in the tunable filter. The actuation voltage reflects the integration performance of the switch, which contributes to the monolithic microwave integrated circuit (MMIC) implementation.

To date, many RF MEMS switches with a high capacitance ratio have been proposed with a large gap between beam and signal transmission line [6,7], a relatively thinner dielectric layer [8] or high dielectric constant materials [9,10,11,12] including HfO_2_ (*ε_r_* = 20), STO (*ε_r_* = 30–120), Ta_2_O_5_ (*ε_r_* = 32). However, the mechanical property of an RF MEMS switch is altered when the height of the beam is changed. In capacitive MEMS switches when the pull-down voltage is applied and the beam comes in contact with the dielectric, charge carriers are injected and then trapped in the dielectric layer, which leads to a change in the pull-down voltage or in the phenomenon of stiction which seriously limits the functionality of devices [12]. Furthermore, if the dielectric layer is thinner, the reliability of switches will be degraded due to dielectric breakdown. Meanwhile, many researchers have paid efforts to design low spring constants for a low actuation voltage, for example, Ya, M.L., et al. [13] designed a COMS RF MEMS switch with a relatively wide beam to achieve a low pull-down actuation of 3 V, but it cannot be easily released. Yong, Q. X., et al. [14] designed a switch with 14 V pull-down voltage, which can be used on frequency reconfigurable antenna. Li-Ya Ma et al. [15] design a 1.397 μm-thick switch beam with a low actuation voltage switch of 3 V, however, the beam is very thin, which is prone to warpage or fracture.

In this paper, a high capacitance ratio and low actuation voltage RF MEMS switch with a Si_3_N_4_ (*ε_r_* = 7.6) standard dielectric material and a relatively lower gap of 2 μm is designed and fabricated for Ka band RF front-end application, as shown in Figure 1. To reduce the actuation voltage, a low spring constant bending folding beam and bilateral drop-down electrodes are designed in the MEMS switch. The measured results indicated that, for the proposed MEMS switch with a gap of 2 μm, the insertion loss is better than −0.5 dB and the isolation is more than −20 dB from 25 to 35 GHz with an actuation voltage of 15.8 V. From the fitted results, the up-state capacitance is 6.5 fF, down-state capacitance is 4.3 pF, and capacitance ratios is 162.

## 2. Design of the High Capacitance Ratio and Low Actuation Voltage Radio Frequency Micro-Electromechanical Systems (RF MEMS) Switch

### 2.1. Metal-Insulator-Metal (MIM) Fixed Capacitor Design to Improve Switch Capacitance Ratios

A traditional capacitive RF MEMS switches include a coplanar waveguide (CPW) to transmission line, a floating membrane as an actuator, and an electrode to provide a bias voltage. By loading a bias voltage, the switch can switch between on and off states. The up-state capacitance *C_u_*, down-state capacitance *C_d_* and capacitive ratio *C_r_* can be calculated as follows [17]:(1)$$C_{r}=\frac{C_{d}}{C_{u}}=\frac{\frac{\varepsilon_{0}\varepsilon_{r}A}{t_{d}}}{\frac{\varepsilon_{0}A}{g+\frac{t_{d}}{\varepsilon_{r}^{}}}+C_{f}}=\frac{g\varepsilon_{r}}{t_{d}}+1$$
where *ε_r_* and *ε*_0_ are the relative permittivity of the dielectric layer and the permittivity of free space, respectively, *t_d_* is the thickness of the dielectric layer, *g* is the air gap, *C_f_* is the edge capacitance between the membrane and the transmit signals when the switch is in up-state, and it can be omitted in most cases.

The surface roughness of switch beam has a great influence on *C_d_*. When considering roughness, *C_d_* can be calculated by Equation (2):(2)$$C_{d}=\frac{\varepsilon_{0}A}{2}(\frac{1}{t_{d}+\frac{d_{1}}{\varepsilon_{r}}}+\frac{\varepsilon_{r}}{d_{1}}),\begin{array}{cc} & C_{pp}=\frac{\varepsilon_{0}\varepsilon_{r}A}{t_{d}} \end{array}$$
where *d*_1_ is the roughness, as shown in Figure 2, when the roughness is 50 nm, *C_d_* is reduced by 30%. The dielectric thickness and *ε_r_* also have a greater impact on the capacitance.

A simple and effective approach to obtain a high capacitance ratio of a MEMS switch is to connect a shunt capacitor [16,18]. To improve the capacitance ratio without degrading the reliability of the switches, a metal-insulator-metal (*MIM*) fixed capacitors is adopted in the design, as shown in Figure 3. The *MIM* capacitor connects to a metal-air-metal (*MAM*) capacitor in series in up state, and the MAM is changed to resistance *R* in down-state. *C_r_* can be calculated by the following equation:(3)$$\begin{array}{lll} C_{r} & = & \frac{C_{d}}{C_{u}}=\frac{C_{MIM}}{C_{MIM}∥C_{MAM}}=\frac{C_{MIM}}{C_{MAM}}+1\\ & = & \frac{\varepsilon_{r}g}{t_{d}}\frac{A_{MIM}}{A_{MAM}}+1=\frac{\varepsilon_{r}g}{t_{d}}\lambda +1 \end{array}$$
(4)$$C_{MAM}=\frac{\varepsilon_{0}}{g}A_{MAM},C_{MIM}=\frac{\varepsilon_{0}\varepsilon_{r}}{t_{d}}A_{MIM}$$
where *λ* is equal to *A_MIM_/A_MAM_*, other parameters are defined in Equation (1).

In order to compare the influence of MIM capacitor size on the capacitance ratio, simulate a switch based on silicon (*ε_r_* = 11.9). A 0.15 μm-thick Si_3_N_4_ standard dielectric material is used. The CPW line has a size of 60/100/60(*G/S/G*) μm, and the size of the beam is 220 μm width and 500 μm length. For the switch without MIM capacitors, the up-state capacitance, down-state capacitance, and capacitance ratios can be calculated simply [19] as 57.8 fF and 3.18 pF, 55. Figure 4 shows the relationship among the parameter of *w*_1_, *w*_2_, and *C_r_* for the switch with *MIM* capacitors.

It can be seen from the Figure 3 that the MIM capacitor can effectively improve the capacitance ratio of the switch. When *w*_1_ = 10 μm, *C_r_* is increased with the increase of the value of *w*_2_. And when *w*_2_ = 30 μm, *C_r_* is decreased with the increase of the value of *w*_1_. It can be found that *C_r_* = 206 with *g* = 2 μm, *t_d_* = 0.15 μm, *w*_1_ = 10 μm, *w*_2_ = 30 μm.

### 2.2. Low Spring Constant Bending Folding Beam Switch Design and Pull-Down Model

Due to the small amount of deformation, the switch beam can be equivalent to a rigid body, and its mechanical properties can be described by linear elastic coefficients [16]. A simple and effective approach to obtain a low actuation voltage of a MEMS switch is to reduce the spring constant of the beam. This paper designs a switch structure as shown in Figure 4a. In order to reduce the spring constant, the switch uses a bending folding beam, which is fixed on the ground by four anchor points. The switch structure and simplified model are shown in Figure 5.

The pull-down voltage of proposed switch is [18]:(5)$$V_{p}=\sqrt{\frac{8k_{total}{\left(g+t_{d}/\varepsilon_{r}\right)}^{3}}{27\varepsilon_{0}A_{total}}}$$
(6)$$A_{total}=2(S_{e}-N\pi {\left(R_{i}\right)}^{2})$$
where *A_total_* is the area of the driving force, *R_i_* is the radius of the circular slots, *N* is the number of circular slots, *S_e_* is the area of driving electrode. Other parameters are the same as previously defined. The spring constants *k_total_* of the beam determines the pull-down voltage, and can be calculated as:(7)$$\begin{array}{c} k=F/\delta \end{array},\begin{array}{cc} & \delta =(\partial V_{\varepsilon }) \end{array}/(\partial F)$$
(8)$$V_{\varepsilon }=\sum_{i}^{n}\left(\int_{l}\frac{M_{i}{\left(x\right)}^{2}}{2EI}dx+\int_{l}\frac{T_{i}{\left(x\right)}^{2}}{2GI_{t}}dx\right)$$
where *F* is the applied force, *δ* is the displacement of the structure in the direction of the applied force, and *V_ε_* is the strain energy under the applied force. *M*(*x*) is the bending moment of the structure, *T*(*x*) is the torque, *E* is the elastic modulus of the material, *G* is shear modulus, *I* = *wt*^3^/12 is the moment of inertia of the section, and *I_t_* = 0.3 *wt*^3^ is the polar moment of inertia.

As shown in Figure 6a, in order to calculate the spring constants of the supporting beam, the equivalent mechanical spring structure can be divided into **① ② ③ ④** four parts, one end is specified and the other end receives a force *F* perpendicular to the surface of the paper. Then the strain energy of each part are calculated in Table 1.
(9)$$\begin{array}{cc} \delta & =\frac{\partial (V_{\varepsilon 1}+V_{\varepsilon 2}+V_{\varepsilon 3}+V_{\varepsilon 4})}{\partial F} \end{array}$$
(10)$$\begin{array}{lll} k_{i} & = & \frac{F}{\delta }\\ & = & \frac{1}{\frac{\left(-7a_{1}^{3}+a_{2}^{3}+{\left(a_{1}+a_{3}\right)}^{3}+(a_{4}^{3}+a_{5}^{3})\right)}{12EI}+\frac{\left(a_{1}^{2}a_{2}+a_{2}^{2}a_{3}+{\left(a_{1}+a_{3}\right)}^{3}(a_{4}+a_{5})\right)}{GI_{t}}} \end{array}$$

The switch is equivalent to the parallel connection of four mechanical spring as shown in Figure 6a, the total spring constant of the switch can be calculated as:(11)$$k_{total}=(k_{1}∥k_{2}∥k_{3}∥k_{4})$$

In this paper, the switch design is based on Au (*E* = 78 Gpa, Poisson ratio *v* = 0.44), length and width of the beam of 500 μm and 220 μm respectively, the gap of 2 μm, and other physical dimensions of the switch shown in Table 2. The *K_total_* is 1.3967 N/m calculated from Equations (6)–(11), and the simulation value is 1.2 N/m with the help of finite element analysis software COMSOL in the electromagnetic environment, and as shown in Figure 6b, the calculated values match well with the simulated ones. The voltage can be calculated to 9.2 V and 9 V, from Equation (5).

## 3. Physical Dimensions of the Proposed Switch

Table 2 shows the detailed physical parameters of the proposed switches. In this paper, the switch is based on silicon (*ε_r_* = 11.9), and a 0.15 μm-thick Si_3_N_4_ standard dielectric material is used. Au material is used to make the switch beam.

## 4. Discussion

### 4.1. Fabrication

The fabrication process of the MEMS switch is shown in Figure 7. The switch is fabricated on 400 μm thickness high-resistance silicon, and a layer of SiO_2_ is located above it. Furthermore, photolithograph negative glue was used to photoetching DC bias lines pattern, electron beam evaporation *C_r_* DC bias lines, then stripped photolithograph negative glue, and 0.15 μm-thickness Si_3_N_4_ is deposited on top of the bias lines. Au layer of 0.2 μm thickness, as CPW lines, is sputtered on the substrate, and a layer of Si_3_N_4_ is located on the top of the transmission lines as a dielectric layer. A 2 μm-thick polyimide is used as the sacrificial layer after the thermal curing process. The 1.6 μm-thick anchors and beams are formed by electroplating technology. The sacrificial layer is released using supercritical dry release. Figure 8 show the micrograph of the fabricated MEMS switch.

The capacitance ratio of the switched capacitor is heavily influenced by the roughness of the capacitive area of the beam. In order to decrease surface roughness of polyimide, this paper proposes chemical mechanical polishing (CMP) with the condition of fine polishing pad, strong alkali (PH10), high temperature (60 °C), and speed 63/58. Figure 9a presents the thickness of the PI vary across the wafer before and after CMP. The roughness of polyimide before CMP is 300–500 nm. It can be found that surface roughness of real contact areas for the beam is controlled within 10 nm that can effectively ensure the capacitance ratio.

### 4.2. Measurement and Analysis

The measurements were carried out in an unpackaged and standard laboratory environment. The vector Network Analyzer (R&S ZVA50, Rohde and Schwarz, Munich, Germany) is used to measure the S (S11 and S21) parameters of the device. Two gold ACP-A-GSG-150 probes (Cascade Microtech, Beaverton, OR, USA) are used to contact the two ends of the device, and the device is placed on a probe table (Cascade Summit 11000B-M, Cascade Microtech, Beaverton, OR, USA). The sweep frequency is from 10 MHz–40 GHz. Simulating the proposed switch through the HFSS software.

As shown in Figure 10, S_11_ and S_12_ of the measured and simulated results are well matched. The measured results show that the return loss is better than −1 dB at 30 GHz in up stated condition, and it is more than −10 dB from 25 GHz to 35 GHz in down stated condition. The insertion loss is better than −0.5 dB at 30 GHz, and the isolation is more than −20 dB from 25 GHz to 35 GHz with an actuation voltage of 15.8 V. The maximum isolation is achieved −39.2 dB at 31.3 GHz.

The measurement results show that the central frequency point and actuation voltage of the fabricated switch are higher than the simulation and theoretical calculation values, and these errors may be generated from the following two aspects: the switch beam warpage caused by stress gradient [17] leads to a larger gap between the beam and the electrodes, which increases the actuation voltage, and changes the downstate capacitance due to insufficient contact, as shown in Figure 11; due to the residual stress during the fabrication process, the spring constants of the beam is increased, which leads to a larger actuation voltage. As shown in Figure 12, when the load is applied on both sides of the beam, the spring constants are composed of stiffness and residual stress, and total spring constant of the switch can be calculated as [17]:(12)$$K_{total}^{}=K_{s}+K_{r}=32Ew{\left(t/l\right)}^{3}\frac{1}{(x/l){\left(1-(x/l)\right)}^{2}}+4\sigma (1-\nu )w(t/l)\frac{1}{1-(x/l)}$$
where *K_s_*, *K_r_* are the spring constants caused by stiffness and residual stress respectively, *x* is the position of the load, *σ* is the residual stress. When *x* = 2*l*/3, *σ* = 0 MPa, 40 MPa, 80 MPa, *l* = 200 μm, *t*/*l* = (0.002–0.006), as shown in Figure 12, the total spring constants of the Au (*E* = 78 GPa, *v* = 0.44) beam increases significantly with the increase of residual stress.

*C_u_*, *C_d_*, *L* and *R* can be extracted from the isolation S_12_. The RF MEMS switch and CPW transmission line consist of five parts and can be expressed by the ABCD matrix [19]:(13)$$\left(\begin{array}{cc} A & B\\ C & D \end{array}\right)=M_{1}M_{2}M_{3}M_{2}M_{1}$$
where M_1_ and M_2_ represents the CPW transmission line part. M_3_ represents the lumped parameter model of the RF MEMS. They are:(14)$$M_{1}=\left(\begin{array}{cc} \cos\theta_{1} & Z_{1}\tan\theta_{1}\\ j\frac{1}{Z_{1}}\sin\theta_{1} & \cos\theta_{1} \end{array}\right)M_{2}=\left(\begin{array}{cc} \cos\theta_{2} & Z_{2}\tan\theta_{2}\\ j\frac{1}{Z_{2}}\sin\theta_{2} & \cos\theta_{2} \end{array}\right)$$
$$M_{3}=\left(\begin{array}{cc} 1 & 0\\ Y_{3} & 1 \end{array}\right)$$
$$Y_{3}(down-state)={\left[{\left(j\omega C_{d}\right)}^{-1}+j\omega L+R_{s}\right]}^{-1}={\left[{\left(j\omega C_{MIM}\right)}^{-1}+j\omega L+R_{s}\right]}^{-1}$$
$$Y_{3}(up-state)={\left[{\left(j\omega C_{u}\right)}^{-1}+j\omega L+R_{s}\right]}^{-1}={\left[{\left(j\omega C_{MIM}∥C_{MAM}\right)}^{-1}+j\omega L+R_{s}\right]}^{-1}$$
where (*Z*_1_, *θ*_1_) and (*Z*_2_, *θ*_2_) are the characteristic impedances and the electrical lengths of the transmission lines with different sizes, *Z*_1_ = 52.8 Ω, *θ*_1_ = 7.65°, *Z*_2_ = 95.2 Ω, *θ*_1_ = 3.82° at 30 GHz, the resistance of *MAM* is ignored, *S*_21_ is:(15)$$S_{21}=\frac{2}{A+B/Z_{0}+CZ_{0}+D}$$

From Equations (12)–(15), for the proposed switch, *C_u_* = 26.5 fF, *C_d_* = 4.3 pF, *C_r_* = 162, the MIM capacitors can improve the on/off capacitance ratio.

Compared with other capacitive RF MEMS switch, as shown in Table 3, the proposed switch in this work shows a lower actuation voltage and a significant increase of capacitance ration.

## 5. Conclusions

A high capacitance ratio and low actuation voltage RF MEMS switch is designed and fabricated for Ka band RF front-ends application in this paper. A metal-insulator-metal (MIM) capacitor is designed on the signal line to improve the capacitance ratio, which will not degrade the switch reliability. The switch has a 1.6 μm-thick low spring constant bending folding beam and bilateral drop-down electrodes, which can greatly reduce the spring constant, and the actuation voltage is as low as 15.8 V. From the fitted results, the up-state capacitance is 6.5 fF, down-state capacitance is 4.3 pF, and capacitance ratios is 162. The measured results indicated that the insertion loss was better than −0.5 dB and the isolation was more than −20 dB from 25 to 35 GHz. This paper has several limitations which should be improved in future research: first, the beam was treated as a rigid body, which is inaccurate for large-size thin plates. Non-linearity of beam elastic coefficient at micro scale should be considered; secondly, the residual stress and warpage were cited as a cause for difference in the measured results and simulation, and they should be taken into account in the theoretical model.

## Figures and Tables

**Figure 1 micromachines-13-00037-f001:**
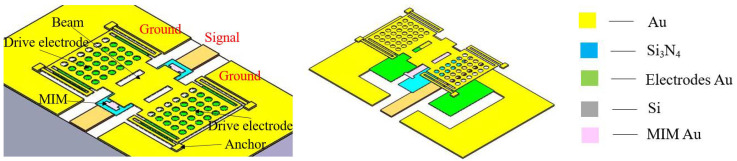
Switch diagram. Adapted with permission from ref. [16]. Copyright 2020, Springer Nature.

**Figure 2 micromachines-13-00037-f002:**
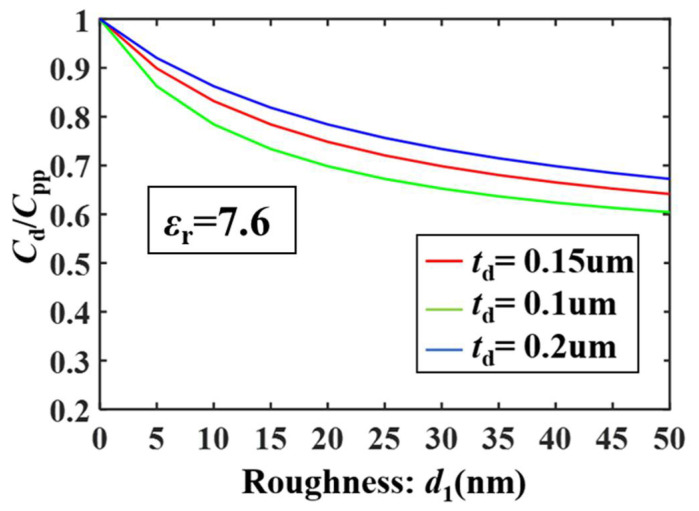
Simulation curves of roughness and *C_d_*.

**Figure 3 micromachines-13-00037-f003:**
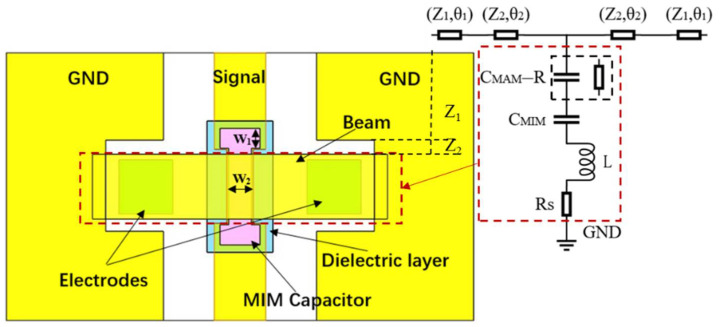
The metal-insulator-metal (MIM) capacitor switch radio frequency (RF) model. Adapted with permission from ref. [16]. Copyright 2020, Springer Nature.

**Figure 4 micromachines-13-00037-f004:**
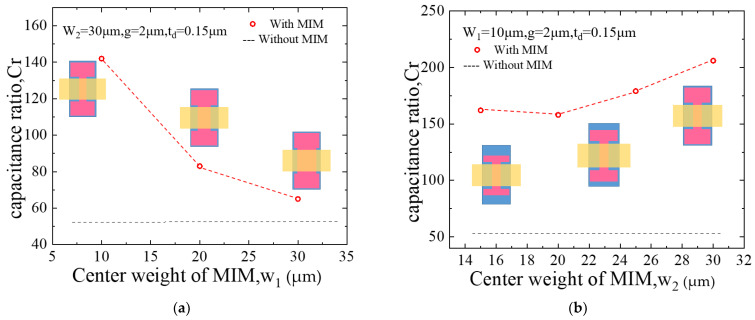
*C_r_* with and without MIM. (**a**) the relationship between *C_r_* and *w*_1_. (**b**) the relationship between *C_r_* and *w*_2_ [16].

**Figure 5 micromachines-13-00037-f005:**
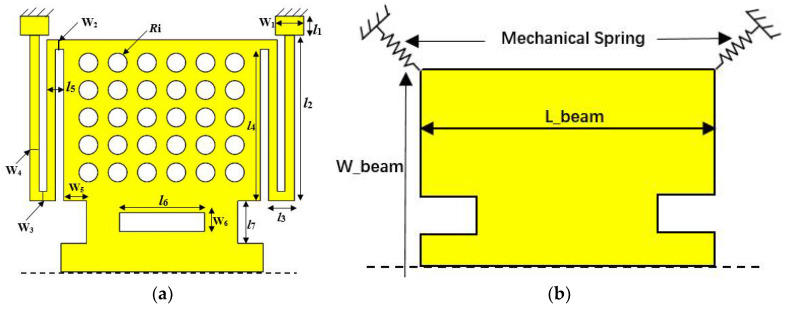
(**a**) Detailed size of the switch; (**b**) equivalent model of switch [16].

**Figure 6 micromachines-13-00037-f006:**
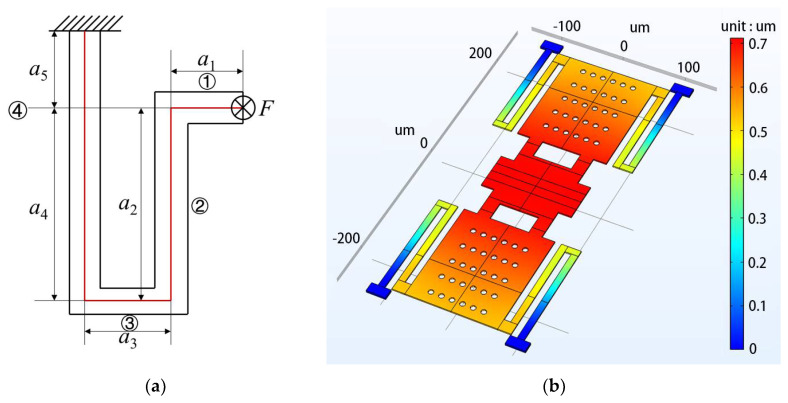
(**a**) Equivalent mechanical spring model; (**b**) switch beam simulation diagram.

**Figure 7 micromachines-13-00037-f007:**
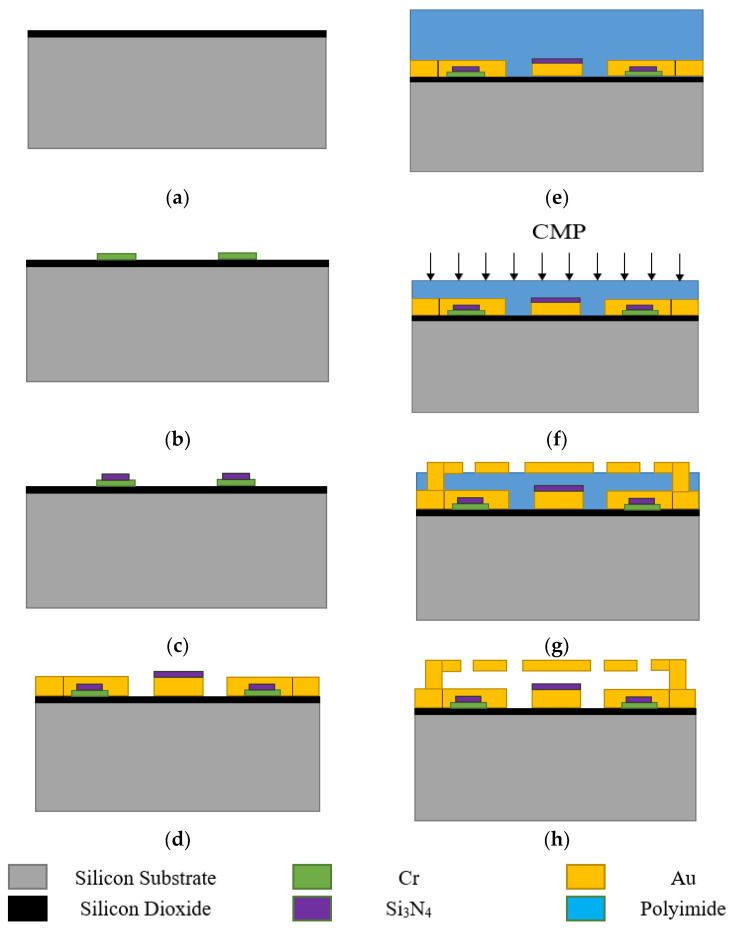
The process of the device: (**a**) silicon substrate after oxidation; (**b**) after patterning of the high-resistance direct current (DC)-bias line; (**c**) after patterning of the Si_3_N_4_ dielectric layer; (**d**) after patterning of coplanar waveguide (CPW) lines; (**e**) spinning of polyimide to form sacrificial layer; (**f**) Chemical mechanical polishing (CMP) to decrease surface roughness of polyimide; (**g**) patterning of gold layer to form MEMS beams and anchors; (**h**) removal of the sacrificial layer.

**Figure 8 micromachines-13-00037-f008:**
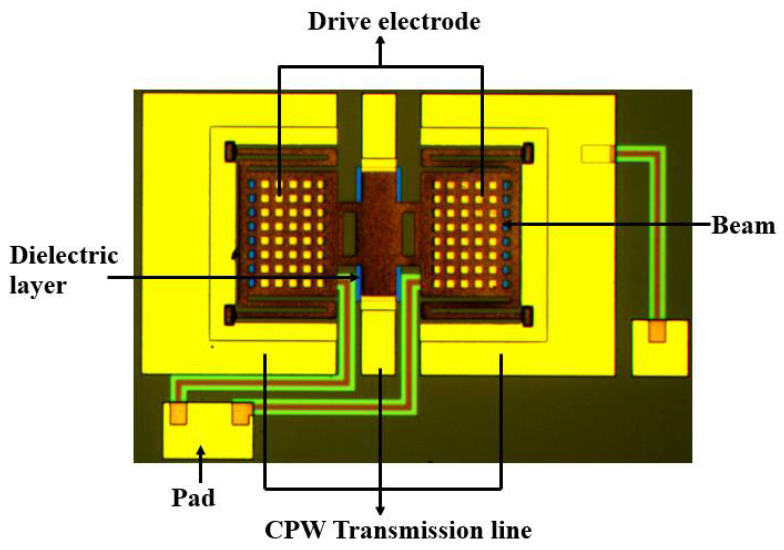
Micrograph of the fabricated MEMS switch.

**Figure 9 micromachines-13-00037-f009:**
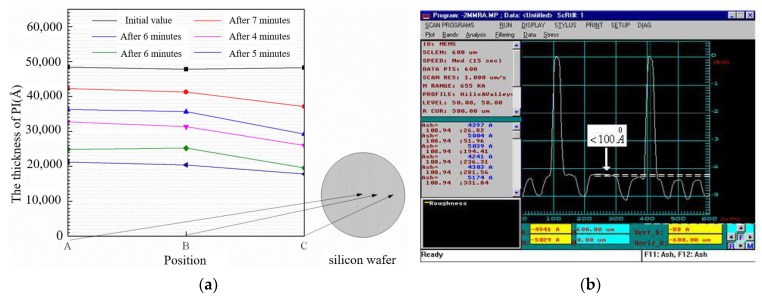
(**a**) PI thickness changes before and after CMP (**b**) Roughness test results.

**Figure 10 micromachines-13-00037-f010:**
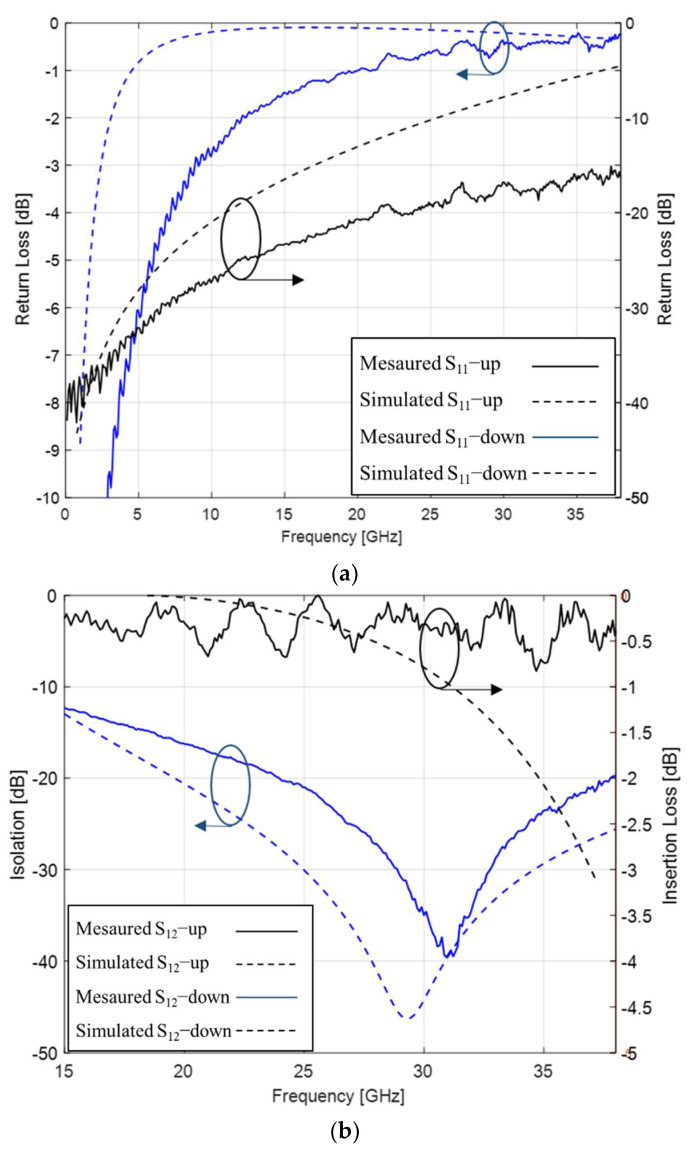
S−parameters of the MEMS switch: (**a**) S11 of the measured and simulated results; (**b**) S12 of the measured and simulated results.

**Figure 11 micromachines-13-00037-f011:**
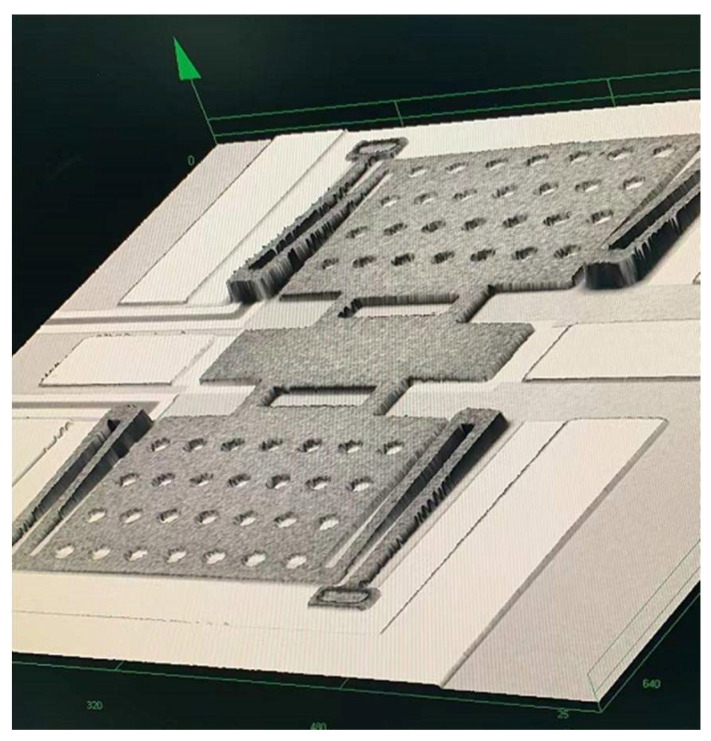
Micrograph of the fabricated MEMS switch.

**Figure 12 micromachines-13-00037-f012:**
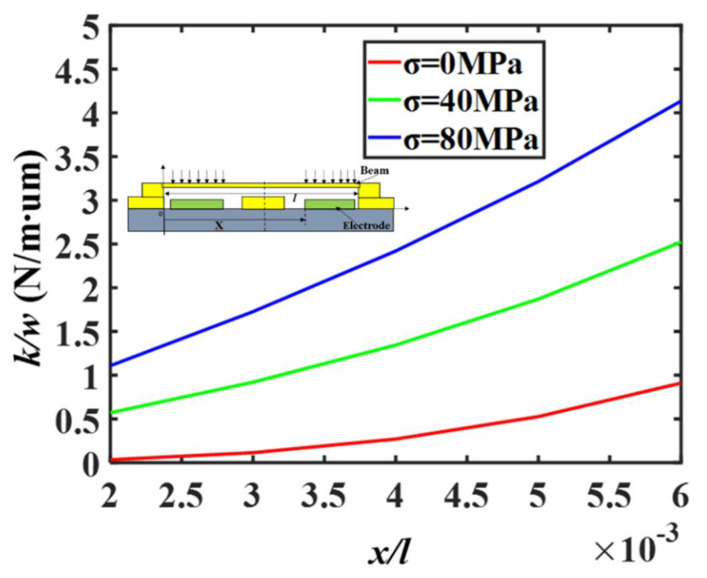
Relationship between elastic coefficient and residual stress.

**Table 1 micromachines-13-00037-t001:** *M*(*x*), *T*(*x*) and *V_ε_* of equivalent mechanical spring structure.

Part	*M*(*x*)	*T*(*x*)	*V_ε_*
①	$M_{1}(x)=Fx_{1}$	$T_{1}(x)=0$	$V_{\varepsilon 1}=\frac{F^{2}a_{1}^{3}}{24EI}$
②	$M_{2}(x)=Fx_{2}$	$T_{2}(x)=Fa_{1}$	$V_{\varepsilon 2}=\frac{F^{2}a_{1}^{3}}{24EI}+\frac{F^{2}a_{1}^{2}a_{2}}{2GI_{t}}$
③	$M_{3}(x)=Fx_{3}$	$T_{3}(x)=Fa_{2}$	$V_{\varepsilon 3}=\frac{F^{2}({\left(a_{1}+a_{3}\right)}^{3}/8-a_{1}^{3})}{6EI}+\frac{F^{2}a_{2}^{2}a_{3}}{2GI_{t}}$
④	$M_{4}(x)=Fx_{4}$	$T_{4}(x)=F(a_{1}+a_{3})$	$V_{\varepsilon 4}=\frac{F^{2}a_{4}^{3}}{24EI}+\frac{F^{2}a_{5}^{3}}{24EI}+\frac{F^{2}{\left(a_{1}+a_{3}\right)}^{2}(a_{4}+a_{5})}{2GI_{t}}$

**Table 2 micromachines-13-00037-t002:** Physical dimensions of the switch [16].

Symbol	Description	Value
l_1_	Length of the anchor	15 μm
l_2_	Length in the mechanical spring	170 μm
l_3_	Length in the mechanical spring	30 μm
l_4_	Length in the rectangular plate	150 μm
l_5_	Length in the mechanical spring	20 μm
l_6_	Length of the rectangular hole	60 μm
l_7_	Length in the rectangular plate	50 μm
l_beam_	Length of the beam	500 μm
w_1_	Width of the anchor	30 μm
w_2_	Width in the mechanical spring	20 μm
w_3_	Width in the mechanical spring	10 μm
w_4_	Width in the mechanical spring	10 μm
w_5_	Width in the rectangular plate	70 μm
w_6_	Width of the rectangular hole	30 μm
w_7_	Width in the MIM capacitor	30 μm
w_8_	Width in the MIM capacitor	10 μm
w_beam_	Width of the beam	220 μm
N	Number of circular slots	56
R_i_	Radius of circular slots	4 μm
t_beam_	Thickness of the beam	1.6 μm
t_d_	Thickness of the dielectric layer	0.15 μm
t_MIM_	Thickness of the MIM capacitor	1 μm
S_e_	Area of actuation electrodes	100 × 150 μm^2^
g	Air gap	2 μm
ε_r_	Relative permittivity of dielectric layer	7.6

**Table 3 micromachines-13-00037-t003:** Comparison of developed capacitive RF-MEMS switches. Adapted with permission from ref. [16]. Copyright 2020, Springer Nature.

Author	*C*_up_ (fF)	*C*_down_ (pF)	*C_r_*	Insertion Loss (dB)	Isolation (dB)	Actuation Voltage (V)
Persano, A. [20]	70	1.32	18.8	<0.8@30 GHz	38@23 GHz	15–20
Park, J. [10]	83	50	600	0.08	42@28 GHz	8
Badia, M.F. [21]	-	1.27	-	0.68@40 GHz	35.8@40 GHz	23.6
Fall, M. [22]	4.8	0.16	33.3	0.3@20 GHz	<1.5@20 GHz	38
Deepak, B. [23]	-	-	-	<0.1@20 GHz	43@9.5 GHz	20
Li-Ya, M. [15]	140	7.31	52	5.65@40 GHz	24.38@40 GHz	3.04
Xu, Y.Q. [14]	-	-	-	0.1@30 GHz	26@30 GHz	14
Ke, H. [24]	54	20.8	385	<0.5@40 GHz	34@10 GHz	21
Fouladi [7]	23	2.1	91	0.98@20 GHz	17.9@20 GHz	82
Liu, Y. [25]	40	7	175	1.51@40 GHz	>30@16–36 GHz	35
Yu, A. [26]	34.4	1.3	38	<0.4@DC−40 GHz	27@40 GHz	-
Ziaei, A. [27]	-	-	400	0.1@10 GHz	38@10 GHz	35–40
Rottenberg [28]	18	7.98	459	0.06@1–15 GHz	25@15 GHz	-
Rizk, J. [29]	10	0.27	27	0.25	20@80–110 GHz	30
This paper	26.5	4.3	162	<0.5@30 GHz	39.2@31.3 GHz	15.8

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
