# Peer review of "Design and Fabrication of a Ka Band RF MEMS Switch with High Capacitance Ratio and Low Actuation Voltage"

_micromachines, 2021, doi:10.3390/mi13010037_

Round 1

Reviewer 1 Report

 The work is very interesting and the presentation is good. In this paper, the authors have explained the design, fabrication aspects of RF MEMS Switch for Ka band Applications. The authors have very clearly expressesd all the fabrication process steps and also addressed the performance parameters of the device as per Capacitance ratio and Pull In Voltage.Based on my obersvations; I strongly recommend this work for Publication.

Author Response

The co-authors and I would like to thank you for the time and effort spend making constructive remarks, which has significantly raised the quality of the manuscript and has enabled us to improve the manuscript. Thanks again for your help.

Reviewer 2 Report

  • Use consistent reference to Figures
  • Refer to Equations by Equation and not formula
  • Correct some grammar and usage
  • The list of capacitive switch references is much larger than presented. The reading of these other references would highlight many of the gaps so that they could have been addressed upfront.
  • Update the references for consistency and correctness. Some initials are punctuated and others not. Consistency and adherence to the format is important.
  • Capacitance - The main source of capacitance difference highlighted was roughness, but the roughness of the contacting surfaces was not highlighted. What are other sources of capacitance variation, such as dielectric thickness and moving plate curvature
    • How does the dielectric thickness variation for a high-k and low-k material compare, especially for a thin layer
  • What are the charging implications for Si3N4
  • What is MAM? -metal-insulator-metal does not fit?
  • εis the permittivity of free space
  • G is the shear modulus
  • What environment was used for simulations? Electromechanical and RF.
    • In the theoretical model, the beam is treated as a rigid body. Is this true for the simulations? Is this a source of difference. Rigid body for a large thin plate may not be as accurate.
    • The stress and warpage were cited as a cause for difference in the theoretical and simulation. Were these included in the simulation? They should be considered.
    • What is the cause of the warpage? stress-buckling or stress gradient. This can be determined from the simulations.
  • What is SiCr? How is it deposited and patterned? A critical part of the process flow is the patterning of the anchors in the polyimide
  • PI CMP - What was the roughness before and after CMP? What is the improvement in the roughness? How did the thickness of the PI vary across the wafer before and after CMP? How well is the PI thickness controlled? The PI thickness is a significant factor to the 3/2power for Vpi. 
  • What instability is being referred to on line 54?
  • The simulated and measured S-parameters do not agree very well. It would be good to better understand the cause of these differences. It may be necessary to measure the actual geometric parameters to update the model. Were the electrical properties measured for verification. How was the RF modeled and what assumptions were assumed? The RF should match better than this.

  • The RF  measurements are probed results that are inherently challenging. Please describe the calibration and performance of these measurements in better detail. Do you have more simple structures to calibrate the RF measurements?

Author Response

The co-authors and I would like to thank you for the time and effort spend making constructive remarks and useful suggestions, which has significantly raised the quality of the manuscript and has to enable us to improve the manuscript. Each suggested revision and comment was accurately incorporated and considered.  Attached please find the response and revised version.

We would like to thank you again for taking the time to review our manuscript.

Reviewer 3 Report

Overall the paper reads well and design, methodology, and results are properly presented. The reviewer suggests:

1) Please provide some insights/suggestions on how the results in this work can be improved. It would be good to understand what are the current challenges you are trying to overcome for this design - and how you are planning to tackle them.

2) Fig. 5b shows a scale with no measurement unit - I assume it is the displacement in meters, but please fix. Also, since you are using COMSOL, it should be straight forward to create a 3D model and show it in place of a 2D model.

3) In Fig. 1, labels looks stretched. Quality of the figure (and of Fig. 2) can be improved. Please increase the font size of Fig. 3.

4) An SEM image in place of Fig. 10 would significantly increase the quality of the reported results.

Author Response

The co-authors and I would like to thank you for the time and effort spend making constructive remarks and useful suggestions, which has significantly raised the quality of the manuscript and has to enable us to improve the manuscript. Each suggested revision and comment was accurately incorporated and considered. Attached please find the response.
